# HIV-Related Stress Experienced by Newly Diagnosed People Living with HIV in China: A 1-Year Longitudinal Study

**DOI:** 10.3390/ijerph17082681

**Published:** 2020-04-14

**Authors:** Yunxiang Huang, Dan Luo, Xi Chen, Dexing Zhang, Zhulin Huang, Shuiyuan Xiao

**Affiliations:** 1Department of Social Medicine and Health Management, Xiangya School of Public Health, Central South University, Changsha 410078, China; huangyunxiang@csu.edu.cn (Y.H.); xiaosy@csu.edu.cn (S.X.); 2Hunan Provincial Center for Disease Prevention and Control, Changsha 410078, China; chenxi161@sohu.com; 3The Jockey Club School of Public Health and Primary Care, Faculty of Medicine, The Chinese University of Hong Kong, Shatin 810016, Hong Kong, China; zhangdxdaisy@cuhk.edu.hk; 4Changsha Center for Disease Prevention and Control, Changsha 410078, China; zlh_cscdc@163.com

**Keywords:** HIV-related stress, newly diagnosed people living with HIV, social support, depression, anxiety, China

## Abstract

This study explored the HIV-related stressors that people living with HIV (PLWH) commonly experience and express as stressful at the time of diagnosis and 1 year later. The factors associated with stress levels and whether social support would moderate the negative effects of stress on psychological health (depressive and anxiety symptoms) were also investigated. Newly diagnosed PLWH were consecutively recruited in this study. Participants rated their stress with the HIV/AIDS Stress Scale at baseline and 1 year later. Social support, depression, and anxiety were also self-reported at both time points. There were significant decreases in stress levels 1 year after diagnosis. Stressors regarding confidentiality, disclosure, emotional distress, fear of infecting others, and excessive attention to physical functions were the most problematic at baseline and 1-year follow-up. A younger age, married status, not living alone, less income, presence of HIV symptoms, and lack of social support were associated with higher levels of stress. No stress-buffering effect of social support on depressive and anxiety symptoms was found in this study. Interventions to reduce stress among PLWH should take into consideration the following priority stressors: confidentiality, discrimination/stigma, serostatus disclosure, distressing emotions, fear of infecting others, and excessive attention to physical functions. More attention should be paid to PLWH with younger age, not living alone, less income, presence of HIV symptoms, and lack of social support.

## 1. Introduction

Highly active antiretroviral therapy (ART) has contributed to dramatic declines in the rate of morbidity and mortality in people living with HIV (PLWH) [1]. HIV infection, rather than a terminal disease, is treated more like a chronic illness [2]. Threats regarding physical survival have to some extent been replaced by the mental health burden related to HIV [3,4]. Being diagnosed with HIV is a traumatic experience for the great majority of individuals, posing considerable stress related to HIV such as HIV-related stigma, disclosure concerns, antiretroviral treatment, and physical changes [5]. PLWH are, therefore, more likely to suffer from greater levels of stress [6].

Moreover, higher levels of stress have been associated with poorer psychological adjustment to HIV diagnosis, as characterized by elevated levels of depressive and anxiety symptoms [7], which have been linked to increased health risk behaviors [8], decreased quality of life [9], and accelerated disease progression [10]. Thus, efforts to reduce stress levels in this population are needed. The stressors experienced by PLWH and factors related to stress levels among this population remain important to identify in order to design effective interventions that help PLWH cope adaptively with this chronic illness.

Much of the research on the stress among PLWH has primarily operationalized stress as generalized stress, namely, negative life-events [11,12,13,14,15]. There are indeed some stressors impacting PLWH that may not be directly linked to HIV and be common in the general population or patients with other chronic disease, such as financial problems and occupational challenges [16]. However, stress specific to HIV may be more concerning for individuals with HIV infection, particularly those who were newly diagnosed as HIV-infected. 

Although some researchers have noticed the limitation of using general life events scales to assess stress among HIV population and have focused on the HIV-specific stress, most of them used qualitative method and were cross-sectional in design [17,18,19,20,21]. A quantitative assessment of stress among PLWH using a disease-specific stress instrument with established validity and reliability is required to elucidate the impact of HIV infection on PLWH. Moreover, a different pattern of stress faced by PLWH may emerge with the development of disease. Following PLWH longitudinally and identifying priority stressors at different stage of disease would be valuable to develop targeted intervention to reduce stress.

Previous studies have identified social support as an important moderator to mitigate the negative psychological consequences of stressful experiences [22,23]. However, diagnosis itself may create potential barriers to the utilization of support for newly diagnosed PLWH [24]. Furthermore, an early study conducted in Canada suggested that the importance of support for HIV population is different at various stages of disease [25]. Whether social support would moderate the negative effects of stressful experiences on psychological health during the early post-diagnosis phase remains unknown.

Therefore, this study aimed to: (1) identify the most salient stressors experienced by PLWH at the diagnosis and 1 year later using a HIV-specific stress scale; (2) investigate the changes and factors associated with stress levels; (3) determine whether social support moderates the relationship between stress and psychological distress (i.e., depressive and anxiety symptoms) in the early stage of the disease, after controlling for the socio-demographic and clinical covariates. 

## 2. Methods 

### 2.1. Participants

From 1 March 2013 to 31 August, 2014, study participants were recruited when they got their HIV-infection diagnosis certification, at the Changsha Center for Disease Control and Prevention, Hunan Province, China. Participants were eligible if they were 18 years or older, had been diagnosed with HIV less than 30 days, and resided in Changsha city for more than 6 months. A self-administered questionnaire including HIV-related stress, social support, depression, anxiety, and socio-demographic and clinical information was completed by eligible participants at baseline and 1-year follow-up. This study was approved by the IEC of the Institute of Clinical Pharmacology at Central South University (CTXY-120033-3), and written informed consent was obtained from each participant before participation.

There were 1267 newly diagnosed PLWH registering in Changsha Center for Disease Control and Prevention in the recruit period. A total of 855 individuals met the inclusion criteria of this study, of which 557 completed baseline survey. Of the 557 participants who completed baseline survey, 410 (73.6%) continued to complete the 1-year follow-up survey. Among those 147 participants who were not followed up, 2 (1.5%) participants dead, 35 (23.8%) transferred to other care centers, 75 (51.0%) refused to participate in the follow-up survey, and 35 (23.8%) could not be contacted.

### 2.2. Measures

#### 2.2.1. Socio-Demographic Variables

The socio-demographic variables of interest for this study included age (in years), gender, marital status (married, divorced/widowed, or single), having children or not, living alone or not, household registration (rural or urban), education (in years), employment status (employed or unemployed), monthly individual income, and sexual orientation (heterosexual, homosexual, or bisexual). 

#### 2.2.2. Clinical Variables

The clinical variables such as CD4 counts, HIV-related symptoms, and ART status after diagnosis were collected. Whether participants had experienced the following symptoms in the month prior to survey was recorded: fever, diarrhea, cough, weight loss, oral thrush, herpes, and tuberculosis. Any other HIV-related symptoms were required to list. The information on CD4 counts and ART initiation was extracted from the Chinese HIV/AIDS Comprehensive Response Information Management System.

#### 2.2.3. HIV-Related Stress

HIV-related stress was measured with the HIV/AIDS Stress Scale (SS-HIV), a HIV-specific measurement of stress, developed by Pakenham et al. in 2002 [26]. The scale is derived from the systematic interviews with PLWH at various stages of the disease, ensuring the stress measures are relevant to the type of stress experienced by PLWH. A culturally adapted and validated Chinese HIV-related Stress Scale (CSS-HIV) was developed by our research group in 2015. Approved by the first author of the SS-HIV (Pakenham), a panel of three bilingual public health experts, who are also trained in psychological fields, translated the original English version of the SS-HIV into simplified Chinese. They also sought clarification from the first author of the SS-HIV with regard to the meaning of multiple items. Additionally, another bilingual medical researcher independently back-translated the finalized Chinese version into English. The panel compared the original English version with the back-translated Chinese version and revised the suboptimal translated words/phrases to ensure consistency with Chinese culture. The CSS-HIV has been shown to be reliable with an overall Cronbach’s α coefficient of 0.906 and a test-retest reliability coefficient of 0.832. The number of HIV-related symptoms, as well as scores on the 9-item Patient Health Questionnaire (PHQ-9) and 7-item Generalized Anxiety Disorder Questionnaire (GAD-7), have also been correlated with the CSS-HIV, indicating acceptable concurrent validity. A detailed description of the cross-culture validation of CSS-HIV is available elsewhere [27]. 

The CSS-HIV consists of 17 items with three types of stress: social stress (6 items), emotional stress (6 items), and instrumental stress (5 items). The social stress was related to difficulties associated with disclosing positive status to others, risk for infecting others, confidentiality, sexual activity, stigma, and discrimination. Emotional stress includes concerns about interpersonal relationship, isolation, bereavement, and distressing emotions. Instrument stress refers to the practical challenges regarding HIV/AIDS management in daily life, including finance, transport, employment, and health care system problems. Participants were asked to check which of 17 events had happened to them during the last month and to rate the intensity on a 5-point Likert-type scale form 0 = “not at all” to 4 = “extremely”.

The responses of “a little bit (1)”, “moderately (2)”, “quite a bit (3)”, and “extremely (4)” suggest that this event is stressful for him/her in the last month. By contrast, a response of “not at all (0)” indicates it is not a stressful event for him/her in the last month. In addition, these ratings are averaged to generate a measure of stress intensity for each stressful event. Scores for three subscales are summed to generate an overall stress score, ranging from 0 to 68. Higher scores indicate higher levels of stress.

#### 2.2.4. Social Support

The Social Support Rating Scale (SSRS) is a 10-item self-administered scale that has been commonly used in different Chinese populations including HIV-infected individuals and has been proved to be valid and reliable [28,29,30,31]. It evaluates 3 types of social support including objective support (3 items), subjective support (4 items), and support utilization (3 items). Objective support assesses the practical support that individual received from a social network. Subjective support is related to the individual’s perception of available support. Support utilization reflects the behaviors that an individual would use when support is needed. The possible range for the overall score of social support is 12 to 66, which were summed for three subscales. Higher scores indicate better social support. 

#### 2.2.5. Depressive Symptoms

The 9-item Patient Health Questionnaire (PHQ-9) was used to assess depressive symptoms in this study [32]. Respondents were asked to report how often they had been bothered by the 9 depressive symptoms in the last 2 weeks. Ratings are made to reflect the degree of severity of depressive symptoms with anchors of 0 = “not at all” to 3 = “nearly every day”. The total score ranges from 0 to 27, with higher scores indicating more severe depressive symptoms.

#### 2.2.6. Anxiety Symptoms 

Anxiety symptoms were assessed with the 7-item Generalized Anxiety Disorder Questionnaire (GAD-7) [33]. Respondents were asked to report the frequency of 7 anxiety symptoms applied to them in the past two weeks. Consistent with PHQ-9 scale, response to each item rates on a 4-point Likert scale from 0 = “not at all” to 3 = “nearly every day”. The total score ranges from 0 to 21, with higher scores indicating more severe anxiety symptoms.

### 2.3. Data Analysis

Participants who had completed two-time point surveys were included in the analysis. First, the descriptive analyses were conducted with percentages, medians, and interquartile ranges (IQR). Second, the sample characteristics among these who completed two-time point surveys and those who were lost to follow-up were compared by Mann–Whitney tests and Chi-square tests, when appropriate. Third, Wilcoxon signed-rank tests were used to test the changes in stress levels between baseline and 1-year follow-up. 

The generalized estimating equation (GEE) method was used to determine the factors associated with stress levels among newly diagnosed PLWH. This method allows us to consider the within-individual correlations between repeated measurements [34]. All variables associated with stress at *p* < 0.2 at bivariate GEE model were entered into multiple analysis. To see whether the influence of variables of interest for this study on stress is different at two-time points, all the significant variables from the multivariate analysis were tested for interaction with time. Only significant interactions were maintained in the final GEE models.

Finally, in order to examine the moderating effect of social support on the association between stress and psychological distress, four separate multivariate GEE models were conducted taking PHQ-9 and GAD-7 scores as the dependent variables. In Model 1, only stress was entered. In Model 2, only social support was entered. In Model 3, stress and social support were entered simultaneously. Model 4 added the interaction term “stress × social support” to Model 3 to test the moderating effect of social support on the association between stress and psychological distress. Each model was adjusted for gender (reference group: male), age (in years), marital status (reference group: married), having children or not (reference group: yes), living alone (reference group: yes), household registration (reference group: urban), education (in years), employment (reference group: yes), income (reference group: ≤ 4000 RMB), sexual orientation (reference group: heterosexual), CD4 counts (reference group: < 200 cells/mm^3^), and presence of symptoms (reference group: yes), as well as ART status at follow-up (reference group: yes). Age and education were included as continuous variables and the remaining as categorical variables. In order to reduce the multicollinearity, the continuous variables including age and education were standardized before the moderation analysis. All analyses were performed using SPSS 24.0 for windows (SPSS Inc., Chicago, IL, USA), and all analyses were considered statistically significant at *p* < 0.05.

## 3. Results

### 3.1. Participants

There were no significant differences in any baseline characteristics between the 410 patients who completed the follow-up survey and 147 patients who did not, except for employment rates (Table 1). Participants lost to follow-up were more likely to be employed than those who completed the follow-up survey (*p* = 0.015). Of the 410 participants who completed two-time point surveys, a majority were men (91.7%), single (61.5%), not lived alone (73.9%), with a median age of 28 years (IQR 24-36). A median of 12 years (IQR 9-15) of education was completed. Nearly one-third of participants were unemployed (29.8%) and more than half (60.9%) reported monthly income of 4000 RMB or less. Only 15.1% of the participants reported CD4 cell counts less than 200 cell/mm^3^, and 37.3% reported presence of symptoms at baseline. Within 1 year after diagnosis, 53.2% of the participants had initiated ART, the median time from ART initiation to follow-up assessment was 6 months (IQR 4-9). Table 1 presents the detailed baseline sample characteristics.

### 3.2. The Frequency and Intensity of HIV-Related Stress

The most frequent stressors experienced by PLWH at diagnosis were confidentiality (93.2%), risk of infecting others (86.9%), distressing emotions (86.3%), physical functions (83.9%), and disclosure concerns (83.7%). The events that were rated as most frequent 1 year after diagnosis were confidentiality (77.6%), disclosures concerns (73.2%), risk of infecting others (71.5%), physical functions (71.2%), and distressing emotions (67.3%). At diagnosis, the disclosure was the fifth most common stressor for participants, while 1 year after diagnosis, it became the second. Participants at diagnosis perceived the distressing emotions as the third most common stressor. However, it was rated as the fifth most common 1 year after diagnosis.

In terms of stress intensity, confidentiality, disclosure, and risk of infection were identified as the three most stressful at diagnosis and 1 year after diagnosis, followed by discrimination/stigma or physical functions. The frequency and intensity of stress are given in Table 2.

### 3.3. Changes in Stress Levels

A statistically significant decline in the overall stress median score was observed after 1 year of being diagnosed (27 (16–40) at baseline to 15 (8–26) at follow-up, *p* < 0.001), and all dimensions of the stress score decreased significantly (*p* < 0.001). Table 3 presents the changes in stress levels among PLWH during the first year after diagnosis.

### 3.4. Factors Associated with Stress Levels

In the multivariate GEE analysis, we found a significant decrease in stress levels at 1-year follow-up survey compared to the baseline survey (β = −13.50, *p* < 0.001). A younger age (*p* = 0.001), married status (*p* = 0.039), not living alone (*p* < 0.001), less income (*p* < 0.001), presence of HIV symptoms (*p* = 0.001), less objective (*p* < 0.001), and subjective support (*p* = 0.017) were associated with higher stress levels. Significant interaction was found between time and objective support (*p* = 0.003). The positive effect of objective support on reducing stress levels diminished over time (β = - 1.31 for baseline; β = −0.54 for 1-year follow-up). Details on the factors associated with stress levels are shown in Table 4.

### 3.5. Moderating Effect of Social Support in the Relationship between Stress and Psychological Distress

After controlling socio-demographic and clinical factors, the main effects of both the stress and social support on depressive (*p* < 0.001 for both) and anxiety symptoms (*p* < 0.001 for both) were significant. Participants with higher levels of stress and lower levels of social support were more likely to experience severe symptoms of depression and anxiety.

However, after entering the stress and social support simultaneously into the model, social support was no longer associated with depressive (*p* = 0.113) and anxiety (*p* = 0.998) symptoms, nor was social support and stress interaction term associated with both depressive (*p* = 0.368) and anxiety (*p* = 0.617) symptoms. Details on the moderating effect of social support on the relationship between stress and psychological distress are shown in Table 5.

## 4. Discussion

This study focused on the HIV-related stress experienced by people who were newly diagnosed with HIV and followed them 1 year later. We found newly diagnosed PLWH are exposed to multiple disease-related stressors and continually struggle with it 1 year after receiving the diagnosis. Prominent challenges for newly diagnosed PLWH were confidentiality, discrimination/stigma, disclosure, distressing emotions in response to diagnosis, fear of infecting others, and excessive attention to physical functions. A younger age, married status, not living alone, less income, presence of HIV symptoms, and lack of objective and subjective social support were associated with higher levels of stress. We did not find a significant buffering effect of social support on the relationship between stress and psychological distress during the first year after diagnosis.

Participants reported worrying constantly about confidentiality related to HIV infection, which was considered as the most frequently and highly stressful event at both time points. In addition, disclosure was found to be another frequently and highly reported stressful event. Actually, the HIV-stigma is the primary cause of stressfulness derived from disclosure and confidentiality, whether it is from external environment or internal cognition [35]. In a meta-analysis, Crawford found that the stigma in HIV is the greatest as compared to many other illnesses such as hepatitis and diabetes [36]. The stigma against HIV is a social process that may be difficult to eliminate in a short-term period. The public awareness of HIV-related issues may be a potential target for intervention.

Ongoing negative emotion was cited as another important challenge at both baseline and 1-year follow-up. The fact that HIV infection is accompanied by multiple psychosocial consequences has been well-documented in previous studies [3,37]. Elevated emotional problems, coupled with poor stress management skills, have been found to be associated with greater risk to poor adherence to ART [38], as well as poor linkage and retention in HIV care [39], which may result in poorer health outcome [10,40]. These results suggest the critical importance of being aware of the psychological health consequences among PLWH. Integrating mental health services into HIV primary care medical appointments may be necessary considering the high burden of mental illness experienced by PLWH [41,42].

Many participants indicated excessively worrying about the development of physical functions. A qualitative study on stressors among youth newly diagnosed with HIV reported that most of them are hyper-vigilant for the presence of physical symptoms after being diagnosed with HIV [17]. Moreover, the potential for infecting others was found to be another major source of stress for newly diagnosed individuals. The misinformation and misconception surrounding the HIV infection may exacerbate the perceived severity and infectivity of this disease. Clinicians can play a critical role by imparting the HIV-related information to PLWH, helping them recognize and manage mild symptoms, working with them to minimize the concerns.

HIV serostatus disclosure to others emerged as a more common stressor 1 year after diagnosis. Having been suffering from such an extremely stressful event for 1 year, PLWH may have a stronger desire to have someone in which they can confide, to cope with this challenging situation [43]. However, they may be afraid of telling others of serostatus due to the fear of potential negative consequences following disclosure such as condemn, isolation, or anger from family or friends [44]. Challenges surrounding HIV serostatus disclosure (i.e., whether disclose serostatus to others, who to disclose, when to disclose, and how to disclose) have been described being particularly stressful to most PLWH in many other studies [17,18,19]. Previous studies also reported that the lack of communicating skills about this sensitive topic can be the barrier to appropriate disclosure [45,46,47]. Therefore, to provide PLWH with disclosure skills may be required in this context, where disclosure may induce a great deal of stress to PLWH.

The overall stress levels among newly diagnosed PLWH decreased 1 year after diagnosis, indicating that the moment of receiving HIV diagnosis is linked to stronger perception of stressfulness and the initial shock of diagnosis, to some extent, wore off over the 1-year period. Quite a few researchers have mentioned the phenomenon of “posttraumatic growth” [48]. It refers to a positive psychological outcome resulting from struggling with a traumatic event. Obviously, receiving a diagnosis of HIV infection is a traumatic event with a strong feeling of stressfulness, whereas the experiences of some form of psychological growth, such as psychosocial adaption, positive belief, and openness to spirituality, have also been observed among many individuals in the aftermath of HIV diagnosis [49]. In addition, Moskowitz et al. proposed that the illness appraisal is central to an adequate understanding of how an individual responds emotionally and adjusts to illness [50]. The shift in the nature of HIV infection from a life-threatening disease to a more manageable chronic illness after the scale-up of ART could be expected to change how PLWH appraise HIV infection. Furthermore, positive illness appraisal has also been closely associated with the positive coping strategies that have been repeatedly taken as an important facilitator, promoting PLWH to adjust to life with HIV diagnosis [51,52]. 

In this study, individuals with a younger age showed higher levels of stress than older individuals. Prior research focusing on the quality of life among PLWH also documented that individuals with older age tend to have better mental health [53]. It is possible that younger individuals need to face more challenges than older such as reproductive decision. Additionally, Katherine et al. mentioned that older individuals with HIV infection may share a more positive perception of their illness than younger counterparts because they are at a later stage in life [53].

Participants with less income were more likely to have higher stress levels in this study. Another cross-sectional randomized trial among HIV patients at 13 clinical sites throughout the US reported that financial problem was the most common stressful event for HIV patients [54]. Economic status is closely associated with the level of resources to which PLWH have access, such as HIV-related knowledge, health insurance, and medical care [55]. Although all HIV-infected individuals have been provided free ART regardless of CD4 counts in China since 2016, the cost of regular medical test is self-financed [56], which may dispose of the patients who have financial problems toward unmanageable and stressful situation. 

Lower levels of stress have been reported by those who were living alone in this study. One possible explanation is that people living alone may be exposed to less social stress such as serostatus disclosure, risk for infecting others, and confidentiality concerns, and thus have relatively lower levels of stress. This may also explain why single individuals had lower levels of stress than married individuals in this study. It appears that living alone is a favorable factor for stress reduction in this study. However, living alone has been also associated with less social support among PLWH [57]. Living alone in the long term may in turn increase stress levels among PLWH due to the lack of social support. 

Consistent with previous studies, symptomatic participants showed higher stress levels than those without symptoms [58,59]. This result suggested that the physical condition and psychological burden among PLWH were closely related. Thus, holistic management of stress among PLWH should take the biological factors into account. No significant association was found between CD4 counts and stress levels in this study. A similar finding was reported in a national community-based study of HIV-infected gay men [12]. Previous studies on the relationship between CD4 counts and stress have been mixed, with some studies reporting individuals with lower CD4 counts had higher levels of stress while others finding no difference [60]. The relationship between CD4 counts and stress warrants further investigation.

This study examined the effects of three types of support on stress levels. After adjusting for other factors, greater subjective support and objective support were significantly associated with lower levels of stress. It suggests perceived support and practical support may be more important for PLWH to contend with stress associated with HIV in the early stage of diagnosis. A significant interaction was also found between time and objective support. The positive effect of objective support on reducing stress levels diminished over time, which is consistent with the viewpoint proposed by Robert et al. that the importance of support for PLWH may be different at various stages of disease [25]. Longer observation on the effect of social support on stress among PLWH at different stages of the disease is needed to deepen our understanding of the role of social support in HIV infection.

In terms of support utilization, association between support utilization and stress levels was not found in this study. During the initial stage of receiving their diagnosis, PLWH may tend to conceal their status from others to prevent the potentially negative reactions as a result of disclosure, which may influence the use of social support among newly diagnosed PLWH [61]. The limited access to social support in most participants after a new diagnosis may explain why the support utilization was not related to stress levels in this study. A low-level utilization of social support in response to HIV diagnosis in the long term may negatively influence mental health by diminishing the overall social support levels among PLWH. In the context of difficulties in seeking support, support services like peer support groups that bring PLWH together to share common stressors related to HIV may be helpful in reducing stress. Health care providers could also be another important source of social support for newly diagnosed PLWH [62].

Contrary to findings in previous studies that social support could moderate the negative consequences of stressful experiences on psychological distress [22,23], we did not find the negative impact of stress on psychological distress was attenuated for individuals with higher levels of social support. Greater social support was associated with fewer symptoms of depression and anxiety when stress was excluded in this model. However, this relationship disappeared when stress was included in the model. This suggests that stress may be a more powerful predictor of psychological adjustment in the early stage of infection and may have overshadowed the contributions that social support confers on depressive and anxiety symptoms. Given the greater stress in response to diagnosis, interventions to screen stressor and reduce stress levels are urgently needed and would be best to implemented early on or close to the HIV diagnosis.

Several limitations should be documented in this study. First, the sample in this study was predominantly consisted of men, with a limited number of other groups (such as women) approached. Therefore, these findings may not be generalized to other sub-groups of HIV population. Future research on the stress among PLWH should consider diverse subgroups of HIV population such as intravenous drug users, pregnant women, and youth living with HIV. Second, additional factors that may be correlated to stress among PLWH such as coping style, stigma, and resilience were not investigated in this study. These variables should be included in future studies. Third, a proportion of people completed the baseline survey but dropped out of the follow-up survey. The results of this study may be affected by attrition bias. In addition, the follow-up period is only 1 year, which is a relatively short period in terms of the whole trajectory of HIV disease. The primary stressors experienced by PLWH may change with the progression of the illness. Longer observation throughout the whole trajectory of HIV infection would be valuable to identify the major stressors that HIV-infected individuals may experience across different stages of the disease. 

## 5. Conclusions

The results from our study may have important implication for healthcare providers and policymakers to develop more targeted and effective interventions on reducing stress among PLWH. The confidentiality and discrimination concerns related to HIV, difficulty in serostatus disclosure, distressing emotions in response to diagnosis, fear of infecting others, and excessive attention to physical functions were presumed to be the primary concern of newly diagnosed individuals. These challenges remained for the primary stressors that individuals struggled with 1 year after being diagnosed. Future stress management should take these stressors into account. Special attention should be given to PLWH with younger age, not living alone, less income, presence of HIV symptoms, and lack of social support.

## Figures and Tables

**Table 1 ijerph-17-02681-t001:** The sample characteristics at baseline.

Characteristics	Baseline Total *n* = 557 (%)	1-Year Follow-up		
		Complete *n* = 410 (%)	Drop ou T*n* = 147 (%)	*p* Value
Gender				
Male	515 (92.5)	376 (91.7)	139 (94.6)	0.362 ^a^
Female	42 (7.5)	34 (8.3)	8 (4.5)	
Age, median (IQR)	28 (27–37)	28 (24–36)	29 (24–38)	0.538 ^b^
Marital status				
Married	139 (25.0)	110 (26.8)	29 (19.7)	0.159 ^a^
Divorced/widowed	71 (12.7)	48 (11.7)	23 (15.6)	
Single	347 (62.3)	252 (61.5)	95 (64.6)	
Having children				
Yes	167 (30.0)	123 (30.0)	44 (29.9)	0.988 ^a^
No	390 (70.0)	287 (70.0)	103 (70.1)	
Living alone				
Yes	158 (28.4)	107 (26.1)	51 (34.7)	0.055 ^a^
No	399 (71.6)	303 (73.9)	96 (65.3)	
Household registration				
Rural	274 (49.2)	200 (48.8)	74 (50.3)	0.746 ^a^
Urban	283 (50.8)	210 (51.2)	73 (49.7)	
Education, median (IQR)	10 (12–16)	12 (9–15)	12 (10–16)	
Employment				
Yes	391 (70.2)	276 (67.3)	115 (78.9)	0.015 ^a^
No	166 (29.8)	134 (32.7)	32 (21.8)	
Monthly income (RMB)				
≤4000	339 (60.9)	259 (63.2)	80 (54.4)	0.062 ^a^
>4000	218 (39.1)	151 (36.8)	67 (45.6)	
Sexual orientation				
Heterosexual	203 (36.4)	152 (37.1)	51 (34.7)	0.874 ^a^
Homosexual	235 (42.2)	171 (41.7)	64 (43.5)	
Bisexual	119 (21.4)	87 (21.2)	32 (21.8)	
CD4 counts, cell/mm^3^	354 (258–466)	357 (254–471)	350 (258–458)	0.930 ^b^
<200	77 (13.8)	62 (15.1)	15 (10.2)	0.229 ^a^
200–500	369 (66.2)	264 (64.4)	105 (71.4)	
>500	111 (19.9)	84 (20.5)	27 (18.4)	
Symptoms				
Yes	199 (35.7)	153 (37.3)	46 (31.3)	0.191 ^a^
No	358 (64.3)	257 (62.7)	101 (68.7)	
HIV/AIDS stress, median (IQR)	21 (12–31)	21 (13–32)	21 (12–30)	0.393 ^b^
Social support, median (IQR)	29 (23–34)	29 (24–34)	30 (22–34)	0.852 ^b^
PHQ-9, median (IQR)	7 (3–13)	8 (3–13)	6 (3–13)	0.305 ^b^
GAD-7, median (IQR)	6 (2–11)	6 (3–11)	6 (2–10)	0.563 ^b^

^a^ Chi-square test; ^b^ Mann–Whitney test.

**Table 2 ijerph-17-02681-t002:** The frequency and intensity of stressors experienced by people living with HIV (PLWH) during the first year after diagnosis.

Stressful Events	Baseline		1-Year Follow-Up	
	Frequency (%)	Intensity (Mean)	Frequency (%)	Intensity (Mean)
Social stress				
Confidentiality concerns	382 (93.2)	2.61	318 (77.6)	1.72
Disclosure concerns	343 (83.7)	2.00	300 (73.2)	1.44
Physical functions or changes	344 (83.9)	1.82	292 (71.2)	1.19
Discrimination/stigma	323 (78.8)	1.81	258 (62.9)	1.22
Risk of infecting others	356 (86.9)	1.96	293 (71.5)	1.35
Sexual difficulties	271 (66.1)	1.28	242 (59.0)	0.98
Emotional stress				
Relationship problems	174 (47.3)	0.8	161 (39.3)	0.57
Distressing emotions	354 (86.3)	1.62	276 (67.3)	0.91
Suicidal thoughts/attempts	151 (36.8)	0.61	79 (19.3)	0.26
Boredom	263 (64.1)	1.15	186 (45.4)	0.64
Isolation	229 (55.9)	1.01	157 (38.3)	0.55
Grief/bereavement	296 (72.2)	1.44	222 (54.1)	0.83
Instrumental stress				
Non-HIV-related health care	210 (51.2)	0.90	188 (45.9)	0.75
HIV-related treatment	241 (58.5)	1.12	196 (47.8)	0.75
Financial problems	194 (48.0)	0.84	130 (31.7)	0.48
Transport problems	135 (12.9)	0.51	98 (23.9)	0.34
Employment problems	284 (69.3)	1.40	208 (50.7)	0.83

**Table 3 ijerph-17-02681-t003:** The changes in stress levels among PLWH during the first year after diagnosis.

Stress Levels, Median (IQR)	Baseline (*n* = 410)	1-Year Follow-up (*n* = 410)	*p* Value
HIV/AIDS stress	21 (13–32)	13 (6–20)	<0.001
Social stress	11 (7–16)	7 (4–11)	<0.001
Emotional stress	6 (3–10)	3 (1–5)	<0.001
Instrumental stress	4 (1–7)	2 (0–5)	<0.001

**Table 4 ijerph-17-02681-t004:** Factors associated with stress levels among newly diagnosed PLWH.

Characteristics	Univariate		Multivariate	
	β Coefficient (95% CI)	*p* Value	β Coefficient (95% CI)	*p* Value
Time				
Baseline	Ref		Ref	
Follow-up	−8.08 (−9.36 to −6.80)	<0.001	−13.50 (−17.63 to −9.36)	<0.001
Gender				
Male	Ref		Ref	
Female	−0.28 (−3.65 to 3.09)	0.872	−	−
Age	−0.13 (−0.22 to −0.04)	0.004	−0.22 (−0.36 to −0.09)	0.001
Marital status				
Married	Ref		Ref	
Divorced/widowed	3.31 (0.09 to 6.54)	0.044	2.10 (−0.90 to 5.09)	0.170
Single	2.27 (−0.08 to 4.61)	0.058	−3.14 (−6.13 to −0.16)	0.039
Having children				
Yes	Ref		Ref	
No	1.10 (−1.06 to 3.27)	0.318	−	−
Living alone				
Yes	Ref		Ref	
No	1.97 (0.04 to 3.90)	0.046	3.86 (1.99 to 5.73)	<0.001
Household registration				
Urban	Ref		Ref	
Rural	−1.89 (−3.87 to 0.09)	0.061	−1.07 (−3.02 to 0.89)	0.286
Education	0.24 (−0.05 to 0.53)	0.099	0.30 (−0.01 to 0.60)	0.057
Employment				
Yes	Ref		Ref	
No	0.67 (−1.35 to 2.70)	0.514	−	−
Monthly income (RMB)				
≤4000	Ref		Ref	
>4000	−3.69 (−5.55 to −1.83)	<0.001	−3.91 (−5.69 to −2.13)	<0.001
Sexual orientation				
Heterosexual	Ref		Ref	
Homosexual	1.58 (−0.61 to 3.77)	0.157	0.93 (−1.17 to 3.03)	0.386
Bisexual	2.70 (−0.26 to 5.67)	0.074	1.13 (−1.64 to 3.89)	0.424
CD4 count, cells/mm^3^				
<200	Ref		Ref	
200–500	−1.46 (−4.47 to 1.55)	0.341	0.66 (−1.98 to 3.30)	0.623
>500	−3.18 (−6.52 to 0.15)	0.061	−0.52 (−3.46 to 2.41)	0.726
Symptoms				
Yes	Ref		Ref	
No	−6.34 (−8.31 to −4.37)	<0.001	−4.34 (−6.14 to −2.53)	0.001
ART				
Yes	Ref		Ref	
No	5.97 (4.33 to 7.61)	<0.001	0.87 (−1.17 to 2.91)	0.401
Social support				
Subjective support	−0.41 (−0.55 to −0.26)	<0.001	−0.19 (−0.35 to −0.03)	0.017
Objective support	−0.61 (−0.88 to −0.34)	<0.001	−1.31 (−1.69 to −0.93)	<0.001
Support utilization	−0.80 (−1.28 to −0.32)	0.001	−0.32 (−0.88 to 0.24)	0.262
Time × objective support	−	−	0.77 (0.26 to 1.27)	0.003

Ref: reference group.

**Table 5 ijerph-17-02681-t005:** The moderating effect of social support on the relationship between stress and psychological distress *.

Variable	Depressive Symptoms		Anxiety Symptom	
	β Coefficient (95% CI)	*p* Value	β Coefficient (95% CI)	*p* Value
Model 1				
Stress	0.66 (0.60 to 0.72)	<0.001	0.28 (0.25 to 0.30)	<0.001
Model 2				
Social support	−0.20 (−0.27 to −0.13)	<0.001	−0.10 (−0.15 to −0.06)	<0.001
Model 3				
Stress	0.65 (0.59 to 0.71)	<0.001	0.28 (0.25 to 0.30)	<0.001
Social support	−0.03 (−0.09 to 0.10)	0.113	−0.0004 (−0.03 to 0.03)	0.998
Model 4				
Stress	0.64 (0.58 to 0.70)	<0.001	0.66 (0.60 to 0.72)	<0.001
Social support	−0.05 (−0.10 to 0.01)	0.107	−0.002 (−0.06 to 0.06)	0.939
Stress × social support	−0.02 (−0.08 to −0.03)	0.368	−0.01 (−0.07 to 0.04)	0.617

* All models were adjusted for gender, age, marital status, having children or not, living alone or not, household registration, education, employment, income, sexual orientation, CD4 counts, symptoms, and ART status.

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
