# Peer review of "HIV-Related Stress Experienced by Newly Diagnosed People Living with HIV in China: A 1-Year Longitudinal Study"

_ijerph, 2020, doi:10.3390/ijerph17082681_

Round 1
Reviewer 1 Report
Greetings Author(s),
Overall, this is a worthwhile manuscript. This study investigates a sample of individuals 18 years or older and were newly diagnosed with HIV; following them for one year. The investigation sought to identify the most salient stressors experienced by PLWH at the diagnosis and 1 year later using a HIV-specific stress scale; (2) investigate the changes and factors associated with stress levels; (3) determine whether social support moderates the relationship between stress and psychological distress (i.e. depressive and anxiety symptoms) in the early stage of the disease, after controlling for the socio-demographic and clinical covariates. The approach could be strengthened by a framework that utilizes constructs of social support, stress, coping, depression, anxiety, etc. The authors may consider frameworks such as the: Andersen Behavioral Model of Health Services Use or focus on stress/coping more directly employing constructs of the Transactional Model of Stress & Coping. The research design and methods are appropriate. Tables are supportive of the findings related in the manuscript. The conclusions drawn from the study are supported by the data as detailed in the paper. Articles mentioned in the discussion may also add more depth to the background section.
Additional considerations:
Line 269 spacing: 1 year , to 1 year,
Line 298 “trail” or trial
Reviewer 2 Report
-The first time you put acronyms in the text (in the summary), you must explain its meaning, for example PLWH.
-The size sample is not specified in Method (Participants), but only in Results.
-The Chinese version of HIV-related Stress Scale 102 (CSS-HIV) only shows feasibility measure, no date about validity is shown nor about the bidirectional translation process (the authors only mention the translation).
-Consider the review of the sentence at line 275: “There have studies pointing out that the lack of...”
Reviewer 3 Report
This study uses multiple measures of HIV-related stress to comprehensively test their hypotheses. The assessment of various aspects of stress and wide range of types of stress (financial versus socially-induced stress) considered in this study is a main strength. The methodology and statistics applied were appropriate and well-described. The results highlight measures of stress that could be used in an intervention, and also describes those at highest risk for longer term HIV-related stress. The results and discussion also appropriately describe how factors identified in this study could be integrated into mental health care for individuals newly diagnosed with HIV.
Comments:
1) While the manuscript is well-written there are some grammatical and typos throughout.
2) You state that "All models were adjusted for gender, age, marital status, living alone, having children or not, household registration, education, employment, income, sexual orientation, CD4, symptoms and ART status." Please include in the methods section how these variables were included, continuous or categorical and what were the reference groups for each?
3) In Table 1, please specify that age was measured in years and if you are presenting the median, range etc. Also, the characteristics in Table 1 should be reformatted. They seem to run together and it is difficult to interpret.
4) In Tables 4 & 5, are you able to transform the results so they may be interpreted as risk ratios? Also, it would be helpful to add in the cell counts to Tables 4 & 5 so that the reader can see the sample sizes behind the associations.
